# Towards a Data-Rich Era: A Bibliometric Analysis of Construction Management from 2000 to 2020

**Shiyao Zhu [1], Dezhi Li [2,*], Jin Zhu [3] and Haibo Feng [4]**

1    School of Transportation and Civil Engineering, Nantong University, Nantong 226001, China
2    School of Civil Engineering, Southeast University, Nanjing 211189, China
3    Civil and Environmental Engineering Department, University of Connecticut, Storrs, CT 06269-3037, USA
4    Department of Wood Science, University of British Columbia, Vancouver, BC V6T 1Z4, Canada
*    Correspondence: njldz@seu.edu.cn

**Abstract:** The rapid development of new technologies has made the acquisition and processing of big data much easier and more accessible to various domains including construction management. This trend has resulted in numerous new technical or management issues in the field, as well as increased research needs. Thus, it is very necessary to invest and assess the past, present, and possible future developments of construction management-related studies. This paper provides a comprehensive bibliometric analysis of the articles published in this field over the previous two decades. The seven most popular research themes were identified and discussed with the data adopted in the related studies, including modern technology, waste management, performance management, risk management, project management, knowledge management, and organization management. Typical research data, research approaches, and future research directions were discussed. Emerging topics such as smart technology, sustainability, resilience, and human factors are suggested to be further explored in the domain. The review conducted in this study can provide some insights into what has been done and what can be achieved in future research in the construction management domain towards a data-rich era.

**Keywords:** construction management; bibliometric analysis; research trend and directions; data-rich era





## 1. Introduction

As a result of various informatics advances in recent years, the world is quickly entering a data-rich era. Various kinds of modern technologies provide opportunities to access large sets of data in different domains, especially civil and construction engineering [1]. Modern, efficient, and reliable construction management helps to provide essential services in our daily life. Construction management involves the management of design, construction, and maintenance of the physical and natural built environment in civil engineering [2], and each one may generate numerous data that are left largely unattended or underused at present. Therefore, the exploration and advancement of construction management-related research have been a continuous effort of both the public and private sectors. It is noted that academic journals play a vital role during this process, as they provide primary context to share research experiences, shape educational programs, and assess academic careers [3]. The structure, content shifts, and patterns of evolution within a discipline can be explored by reviewing and analyzing the articles published in a certain field. Bibliometric analysis has been widely used for reviewing research in many science and engineering fields, such as architecture, geography, automation science, business, food chain, and neuromarketing [4–9]. It is considered to be a common research tool for explaining science-based production and research trends for a given topic [10,11]. It can help to enhance the visual and logic understanding of review findings by clustering and measuring the performance and pertinence of papers with mathematical modeling and algorithms [12,13].

There has been a long history of construction management-related reviews focusing on research methods, techniques, and applications. For example, there are reviews of economics in construction management in the early 1990s [14], reviews of Delphi method used in this field in 2000 [15,16], reviews of construction management methods [17], and reviews of new directions in construction management [18]. These reviews all helped to better understand the research trend in this area. However, the reviews have been studied in an earlier period and are more traditional in their presentation of issues. With entering the information and data-rich era, the research and practices in construction management have changed a lot. With applying big data technology, it has made innovations and improvements in construction management as well as increased new technical or management problems. This growing interest in discovering data-driven technology, new management methods, and smart construction practice in the field of construction management research have led to an increasing number of publications.

Hence, in an effort to dig deeper into the construction management domain, and in order to better understand the existing findings, research gaps, and future directions in this field, a literature review of the publications is very necessary. On the theoretical side, it will help to identify the major research hotspots and provide guidance for academia to conduct future research with the help of data-driven methods and technology. On the practical side, it can provide practitioners with a holistic view of the changes in this field and help them find solutions to adapt to new requirements in the modern era. Therefore, this paper performs a bibliometric analysis of the related studies, aiming to invest and assess the past, present, and possible future developments in the discipline of construction management, taking into account the vast amount of data used in the field.

The paper is organized as follows. After the introduction, the research method is introduced in Section 2. The results of the bibliometric analysis on articles published from 2000 to 2020 are elaborated in Section 3 with an overview of the selected literature, the general research themes identification, and research trend analysis. Section 4 provides further discussions on the data and approaches being adopted and presents the future research directions. Section 5 concludes the paper.

## 2. Materials and Methods

A three-step method was adopted in this study to explore the worldwide trends in construction management scientific production over time, taking into account changes in this field when entering into the data-rich era (Figure 1). First, the data was retrieved from a main database commonly used by researchers. Then, a bibliometric analysis was used to explore the overview of the articles, general research theme, and research trends in an objective and reliable way. Last, in order to show the new changes of construction management in the data-rich era, integrative analysis was conducted to summarize and discuss the data, approaches, and technologies adopted in the selected articles.

### 2.1. Data Collection

As a traditional and comprehensive citation analysis database, Web of Science (WoS) can support longer time citation analysis and high-quality scholar data compared to other search engines, such as Scopus [19]. The database WoS provides a more comprehensive list of construction management journals than comparable databases do, and scholarly metadata (abstracts, titles, keywords, source, and cited references) is available and reliable. All cited references for all publications are fully indexed and searchable in this database, and Citation Alerts, a WoS modular, makes it easier to track citation activity. On 5 February 2021, publications related to construction management were collected from the WoS Core Collection via electronic resources. The keyword "construction management" was searched in the title of the publications in the database with studies written only in English. Therefore, a total of 1393 publications from 2000 to 2020 were found with records including the title, author, abstract, keywords, and references. Then, the type of the publications was set as only for an article. Compared with other forms of literature, articles have complete and

clear research objectives, research contributions, research methods, and research results. Thus, other types of documents, including news, meetings, abstracts, editorial material, and letters were not selected for analysis. Therefore, in this paper, 1188 articles were selected as the literature sample for the follow-up bibliometric analysis.

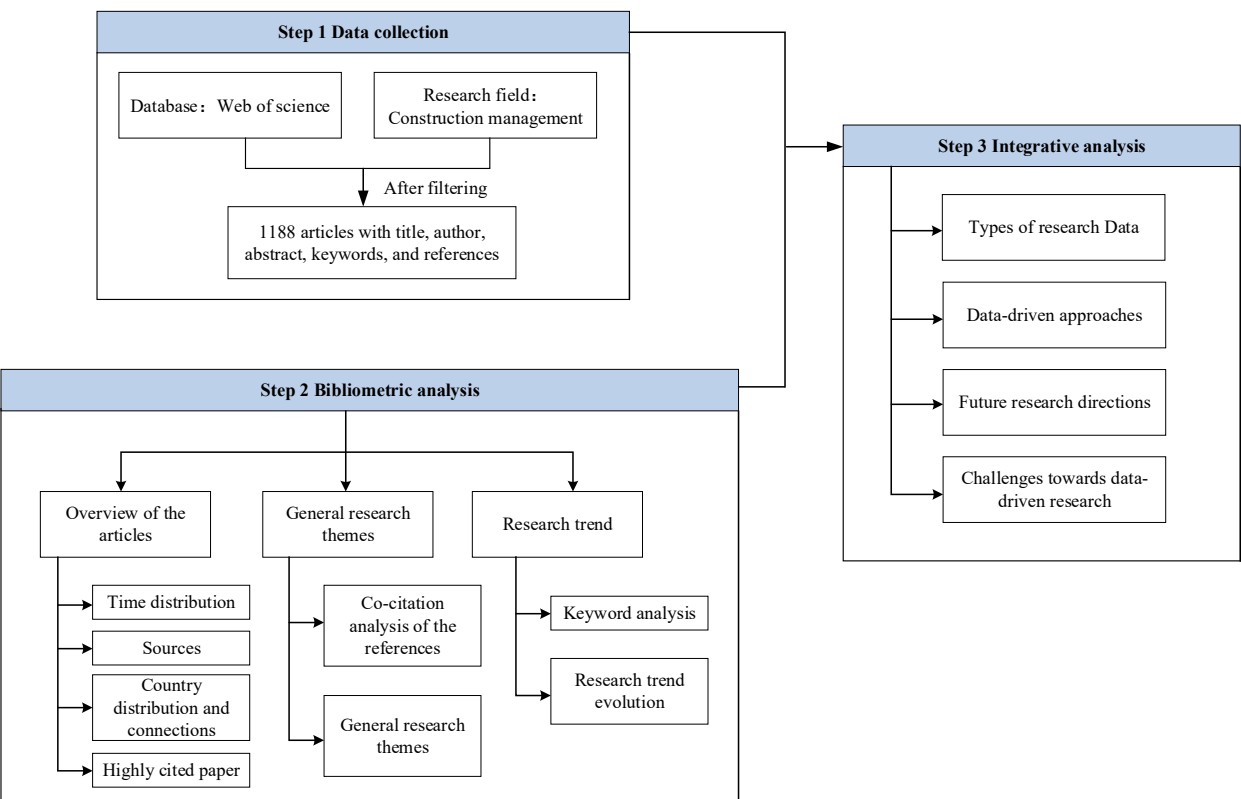

**Figure 1.** Flowchart of the methodology.

### 2.2. Bibliometric Analysis

A bibliometric analysis, which is a quantitative approach for analyzing academic literature using bibliographies to provide the description, evaluation, and monitoring of published research [20,21], was used to analyze the trends in the field of construction management. Every bibliometric method is useful for specific research questions and the most common questions can be answered using bibliometrics for science mapping [22,23]. The bibliometric analysis in this study was designed and performed with the following steps.

First, the overview of the articles was analyzed according to time, source, region, and citation. The time distribution from 2000 to 2020 was illustrated to show the development trend of construction management. The sources of the articles were used in hot journals in this field. The region analysis including country distributions and connections was used to show how the specific works influence a scientific community and social structures. The annual citation structure of the articles was also explored and the highly cited paper was identified.

Second, the co-citation analysis of references was conducted to find research clusters and identify research themes. Co-citation analysis is one of the quantitative techniques capable of enhancing the visual and logical understanding of systematic review findings by clustering and measuring the performance and pertinence of papers with mathematical modeling and algorithms [12]. It is also considered to be a common research tool for explaining science-based production and research trends for a given topic [10]. In a co-citation network, nodes represent cited articles and links represent instances of co-citation, when two articles are cited together by other documents. The co-citation network was developed using *CiteSpace*, which is a versatile toolkit designed for visualizing patterns and

trends in scientific literature [24]. In *CiteSpace*, the cited references are divided into a number of clusters based on their themes and shown in different colors in the co-citation network. The references are closely linked within a cluster in the same color, but loosely connected between different clusters in different colors [24]. It is efficient to define and map the key areas, emerging trends, and interconnections between them with the help of network visualization, spectral clustering, automated cluster labelling, and text summarization in *CiteSpace* [19,24].

Last, with the identified research themes, keyword analysis and research trend evolution were provided to show the research trend. Keywords can help to identify long term research topics within a given field [25]. Keyword analysis is used here as a complement to the co-citation analysis to reveal research trends. The analysis was performed using Bibliometrix R, which is a kind of R tool for constructing and visualizing bibliometric networks and quantitative research [20]. R is one of the most powerful and flexible statistical software environments, providing an open source route to participation. Therefore, R is an integrated suite of software applications for data manipulation, calculation, and graphical display.

With *Bibliometrix R*, a strategic diagram for each period via keyword co-occurrence analysis to show the evolution of the research trends can be generated [20]. The threshold value was five occurrences to be shown in the figure. In Figure 2, the X axis in the strategic diagram presents centrality, which refers to the degree of interaction of one keyword cluster with other clusters, also known as the relevance degree. The Y axis in the strategic diagram presents density, which refers to the internal cohesion of a cluster, also known as the development degree. Thus, these two measures can help to provide the first step for dynamic analysis [26]. Four quadrants were also defined in Figure 2 as motor theme, niche theme, emerging or declining theme, and basic theme. Keyword clusters in Motor Theme present both high centrality and density, which means they are important in the field and have probably been studied over a long period by researchers. Niche Theme shows the topics discussed in these clusters are highly developed and could become a motor theme in the future. The keyword clusters in Emerging or Declining Theme can be defined as edge themes. These themes have not got good development; they may have just emerged or may be about to disappear. The keyword clusters in Basic Theme mainly refer to the important basic concepts in the field.

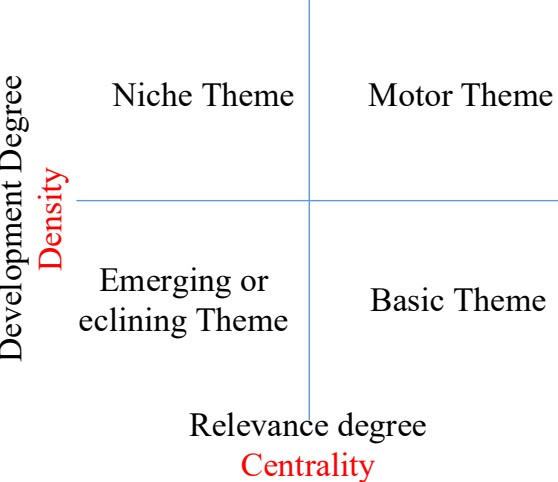

**Figure 2.** Sample of the strategic diagram.

### 2.3. Integrative Analysis

In order to discover the major changes brought by the data-rich era in construction management research, an integrative analysis was conducted with the bibliometric outputs to discuss the type of research data, data-driven approaches, and technologies adopted in the selected articles. Possible research directions were identified and suggestions were

proposed for researchers and practitioners to endeavor to make continuous advancements in construction management theories and practices.

## 3. Bibliometric Analysis Results

### 3.1. Overview of Articles

#### 3.1.1. Time Distribution

The number of articles related to construction management published annually from 2000 to 2020 is shown in Figure 3. It is clear that the articles over the years were not uniform. With regards to articles per year, the samples were split into two periods ensuring a maximum article per year in the First period lower than 50, the threshold which was exceeded in 2009. Accordingly, the First Period (Period 1) was from 2000 to 2009, and the Second Period (Period 2) was from 2010 to 2020.

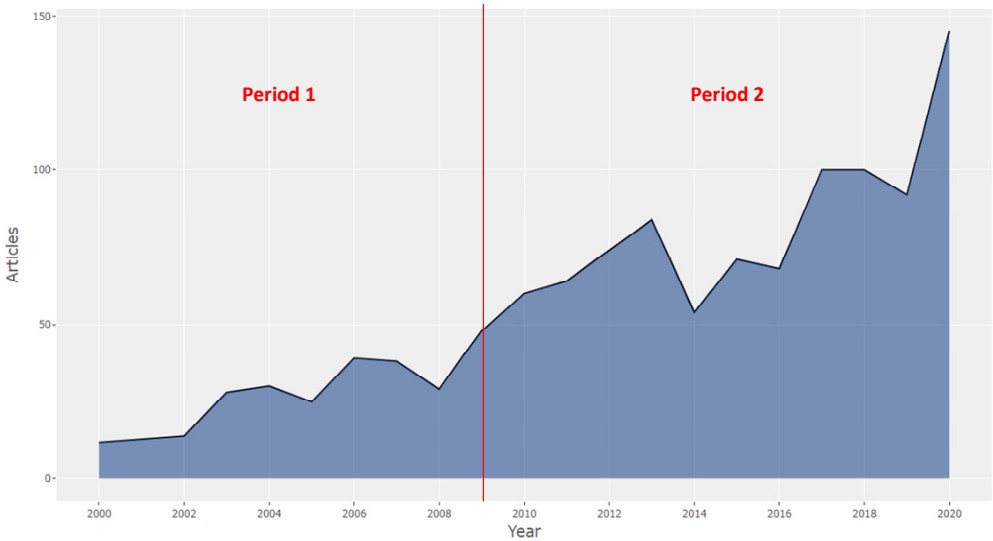

**Figure 3.** Number of articles related to construction management published during 2000–2020 (*generated by Biobliometrix R*).

The main information for each period was also provided in Table 1 with bibliometric analysis. The number of the documents and almost all the main indicators were clearly greater in Period 2, with nearly 3.5 times the number of authors than Period 1. The sources of the documents, as well as the numbers of authors and co-authors per document, were also greater. Nevertheless, the number of documents per author declined in Period 2, and this statistic can be explained when the increase in the collaboration index is taken into account.

**Table 1.** Main information for each period.

| Main Data | Period 1 (2000–2009) | Period 2 (2010–2020) | Total (2000–2020) |
|---|---|---|---|
| Documents | 276 | 912 | 1188 |
| Sources | 125 | 304 | 386 |
| Keywords plus (ID) | 242 | 1499 | 1613 |
| Author's Keywords (DE) | 629 | 2953 | 3370 |
| Average citations per document | 33.33 | 18.02 | 21.58 |
| Authors | 621 | 2200 | 2716 |
| Documents per Author | 0.444 | 0.415 | 0.437 |
| Authors per Document | 2.25 | 2.41 | 2.29 |
| Co-authors per document | 2.67 | 3.18 | 3.06 |
| Collaboration index | 2.52 | 2.6 | 2.48 |

### 3.1.2. Sources

The results of the top 10 productive journals in each period are summarized in Table 2 by their bibliometric indicators. The level of the productivity of the sources changed considerably between the two periods. The total number of the sources in Period 2 was almost 2.4 times that in Period 1.

**Table 2.** Top 10 sources for the publication.

| No. | Period 1 | P | Period 2 | P | Total period | P | H | TC |
|---|---|---|---|---|---|---|---|---|
| 1 | Automation in Construction | 37 | Journal of Construction Engineering and Management | 64 | Automation in Construction | 91 | 40 | 4053 |
| 2 | Journal of Construction Engineering and Management | 24 | Automation in Construction | 54 | Journal of Construction Engineering and Management | 88 | 26 | 2069 |
| 3 | Journal of Construction Engineering and Management-ASCE | 24 | Journal of Management in Engineering | 46 | Journal of Management in Engineering | 59 | 23 | 1557 |
| 4 | Journal of Management in Engineering | 13 | Sustainability | 36 | Sustainability | 36 | 8 | 185 |
| 5 | Journal of Professional Issues in Engineering Education and Practice | 8 | Engineering Construction and Architectural Management | 32 | Engineering Construction and Architectural Management | 32 | 11 | 248 |
| 6 | Building and Environment | 6 | Journal of Civil Engineering and Management | 29 | Journal of Civil Engineering and Management | 32 | 14 | 588 |
| 7 | Waste Management | 6 | Journal of Cleaner production | 29 | Journal of Construction Engineering and Management-ASCE | 32 | 20 | 1154 |
| 8 | Journal of Computing in Civil Engineering | 5 | International Journal of Project Management | 24 | Journal of Professional Issues in Engineering Education and Practice | 31 | 11 | 348 |
| 9 | Resources Conservation and Recycling | 5 | Journal of Professional Issues in Engineering Education and Practice | 23 | Journal of Cleaner Production | 29 | 18 | 973 |
| 10 | Canadian Journal of Civil Engineering | 4 | KSCE Journal of Civil Engineering | 17 | International Journal of Project Management | 26 | 20 | 1506 |

Note: P = production of the articles; H = H index; TC = total citations of the total articles belong to a source.

Considering specific sources, four journals published more than 10 articles in Period 1 and all the top 10 journals in Period 2 published more than 10 articles. The results showed a gradually growing interest in studies related to construction management. In total, *Automation in Construction* published 7.66% of the total publications, followed by *Journal of Construction Engineering and Management* and *Journal of Management in Engineering*, with more than 7.4% and 4.9%, respectively. It is worth noting that in the source growth map (Figure 4), *Sustainability* and *Journal of Cleaner Production* appeared in the high positions of Period 2 and are also the most relevant sources in the total journals. Journals such as *Building and Environment*, *Waste management*, *Resources Conservation and Recycling*, and *Canadian Journal of Civil Engineering* have more articles about construction management in Period 1 only. Other journals such as *Engineering Construction and Architectural Management* evinced a similar trend, with a greater volume of publications about construction management in Period 2.

Considering the total number of citations, *Automation in Construction* was the most cited source with about 2000 citations more than the journal ranked second. *Sustainability* has the fewest number of citations, and its H-index was also ranked the last. In addition to this, other top sources all had an impact greater than 10 in the research field of construction management.

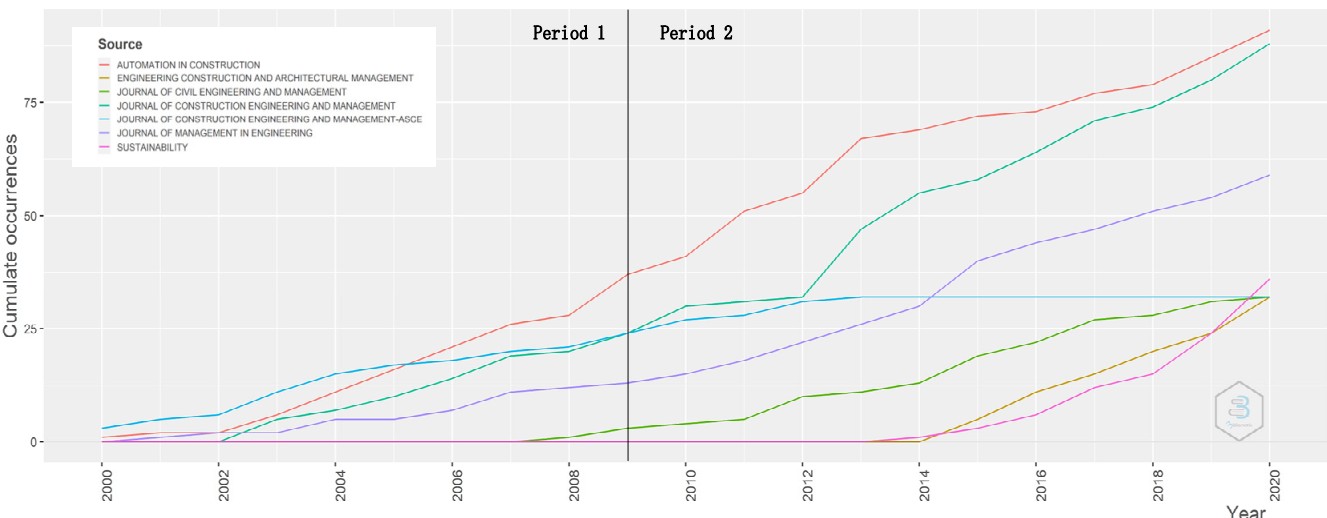

**Figure 4.** Source growth map of the two periods (*generated by Biobliometrix R*).

### 3.1.3. Country Distributions and Connections

The number of articles published and their citations from authors in various countries and institutions can reveal the influence and research progress of these countries and institutions. Seven countries have published more than 100 articles in this field from 2000 to 2020. Among them, China has the highest number of publications (315) and citation times in WoS (7155), accounting for nearly 26.5% of all publications, followed by the USA (225 articles, 3794 citations) and UK (119 articles, 2216 citations). An interesting finding is that researchers from Jordan held the highest average citation rate (95 citations per article), followed by Thailand (61.2 citations per article) and Argentina (61.0 citations per article).

The international country collaboration network was developed with *Bibliomatrix R* as shown in Figure 5. Each node in the figure presents a country/region. The existence of a link between two nodes means the authors from the two countries/regions have a collaboration in one or multiple articles. The thicker the link, the more cooperation between the countries/regions [27]. The most frequent interactions were between China and Australia with 46 collaborative articles, followed by China and the USA with 29. The country that has more links with other countries is identified as the most collaborative country, which is USA in this case.

### 3.1.4. Highly Cited Papers

During the 21 years, 1188 articles published in the field of construction management were cited 25637 times in all sources as reported in WoS with an average citation of 21.58 times/article. The top 10 most cited articles are listed in Table 3. Two indicators were generated for each article: total count of citations and citations per year (CY). CY represents the citation counts per year from paper publication to the end of the collection, which is the year of 2020. This indicator is affected by the citation times as well as the publication year. Therefore, the most cited articles might not have the highest score for count/year. The article with the highest score of CY means it gains attention more quickly than others. All the 10 influential articles have more than 150 citations. Among them, four were published in Period 1, and six in Period 2, which indicates that construction management continues to attract very influential articles during the study period. The top cited article (No. 1) studied the status of safety management in the Chinese construction industry, and identified the importance of the contractor's behavior on safety management [28]. Two articles (No. 2 and 6) focused on the practice of management, one based on the promotion of learning and innovation in organizations [29], and another introduced a new method for sustainable development of project feasibility study [30]. Articles No. 3 and 7 began to discuss green construction and management, including the delivery of cost-effective green buildings [31]

and the challenges faced by project managers who execute green construction projects [32]. Two articles (No. 4 and 9) studied the performance of construction projects and their influence factors [33,34]. Article No.4 is a review paper, which provides a review of studies on stakeholder management in mega construction projects. The last one (No. 10) explored construction supply chain management with new techniques [35]. The data used in this research are qualitatively from literatures or interviews and quantitatively from investigation and questionnaires.

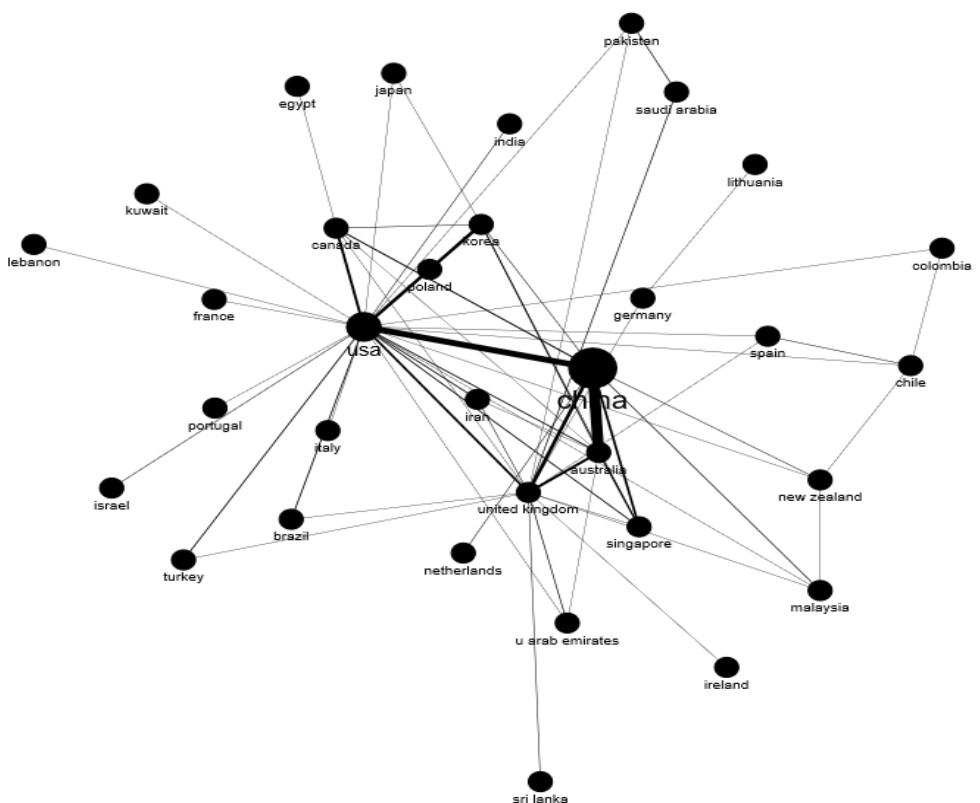

**Figure 5.** Country collaboration network from 2000 to 2020 (generated by *Bibliometrix R*).

*3.2. General Research Themes*

3.2.1. Co-Citation Analysis of the References

Co-citation analysis can help to model and monitor the intellectual structure of scientific specialties over time [36]. In this paper, a co-citation network of cited references is used to reveal the scientific contributions of articles in the field of construction management. As shown in Figure 6, an 851-node and 2674-link hybrid network of co-cited references between 2000 and 2020 was formed. A total of 24 clusters were initially identified from the network. Those clusters with less than 10 articles (#12–15, #17–18, and #20–22) were automatically excluded by *CiteSpace*. Therefore, 15 major clusters were ultimately illustrated in Figure 6.

Two network measurements were used to assess the co-citation network. First is modularity, which measures the structure of the network. Having the value of modularity close to 1 indicates a strong community structure [37]. The second is silhouette value, which measures how similar an article is to its own cluster compared to others. A high value of silhouette indicates that the article is well-matched to its own cluster and poorly matched to neighboring clusters [38]. The clusters shown in Figure 6 have high silhouette scores ranging from 0.907(#10) to 0.998(#23), suggesting these cluster members have high consistency [39]. Overall, the network shown in Figure 5 has a high modularity of 0.8643 and a high mean silhouette value of 0.9391, which suggest that the clustering and network are well-structured and reliable.

Different clusters' detailed information is summarized in Table 4. The label of each cluster is generated by a log-likelihood ration (LLR) test method, which can select the best cluster labels in terms of uniqueness and coverage [24]. In order to overcome the limitation of automatic labeling [24], a manual review of the representative publications in each cluster was used for summarizing the research themes.

**Table 3.** Top 10 most cited articles in all sources.

| No. | Titles | Year | Source | Citations | CY | Research Data Used for Analysis |
|---|---|---|---|---|---|---|
| 1 | Identifying elements of poor construction safety management in China | 2004 | Safety Science | 280 | 15.56 | Qualitative data from literatures and questionnaires |
| 2 | The construction of 'communities of practice' in the management of innovation | 2002 | Management Leaning | 266 | 13.30 | Qualitative and quantitative data from case investigation |
| 3 | Greening project management practices for sustainable construction | 2011 | Journal of Management in Engineering | 220 | 20.00 | Qualitative data from literatures |
| 4 | The effect of relationship management on project performance in construction | 2012 | International Journal of Project Management | 193 | 19.30 | Qualitative and quantitative data from questionnaires |
| 5 | Stakeholder management studies in mega construction projects: A review and future directions | 2015 | International Journal of Project Management | 186 | 26.57 | Qualitative data from literatures |
| 6 | Project feasibility study: the key to successful implementation of sustainable and socially responsible construction management practice | 2010 | Journal of Cleaner Production | 181 | 15.08 | Quantitative data from reports |
| 7 | Project management knowledge and skills for green construction: Overcoming challenges | 2013 | International Journal of Project Management | 170 | 18.89 | Qualitative and quantitative data from literatures and interviews |
| 8 | A fuzzy approach to construction project risk assessment and analysis: construction project risk management system | 2001 | Advances in Engineering Software | 162 | 7.71 | Quantitative data from case investigations |
| 9 | Management's perception of key performance indicators for construction | 2003 | Journal of Construction Engineering and Management | 161 | 8.47 | Quantitative and qualitative data from literatures and questionnaires |
| 10 | Integrating BIM and GIS to improve the visual monitoring of construction supply chain management | 2013 | Automation in Construction | 157 | 17.44 | Quantitative data from case investigations |

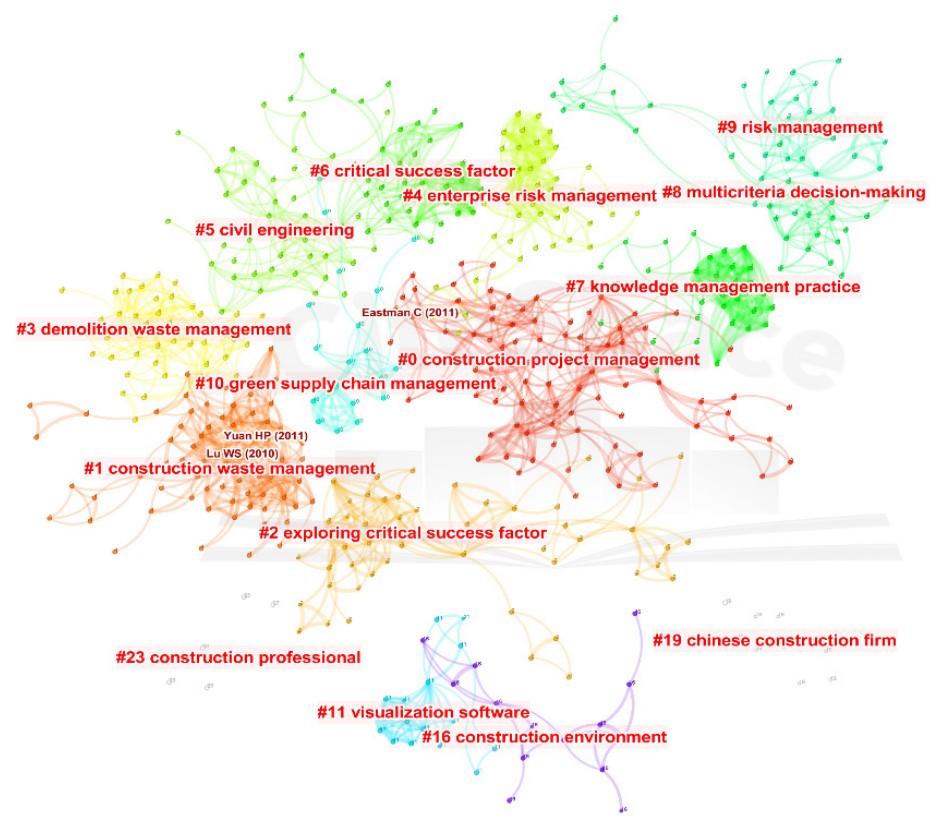

**Figure 6.** Cluster view of co-citation network of cited reference (generated by *Citespace*).

**Table 4.** Fifteen major clusters of co-cited reference.

| # | Size | Silhouette Value | Mean Year | Label (LLR) | Research Themes |
|---|------|------------------|-----------|-------------|-----------------|
| 0 | 90 | 0.921 | 2009 | Construction project management | Modern technology |
| 1 | 70 | 0.939 | 2009 | Construction waste management | Waste management |
| 2 | 54 | 0.944 | 2004 | Exploring critical success factor | Performance management |
| 3 | 44 | 0.922 | 2015 | Demolition waste management | Waste management |
| 4 | 40 | 0.954 | 2010 | Enterprise risk management | Risk management |
| 5 | 40 | 0.907 | 2014 | Civil engineering | Project management |
| 6 | 40 | 0.949 | 2013 | Critical success factor | Performance management |
| 7 | 32 | 0.927 | 2007 | Knowledge management practice | Knowledge management |
| 8 | 30 | 0.974 | 2016 | Multicriteria decision-making | Performance management |
| 9 | 25 | 0.964 | 2003 | Risk management | Risk management |
| 10 | 21 | 0.907 | 2012 | Green supply chain management | Performance management |
| 11 | 16 | 0.988 | 2001 | Visualization software | Modern technology |
| 16 | 13 | 0.938 | 2005 | Construction Environment | Risk management |
| 19 | 9 | 0.997 | 2002 | Chinese construction firm | Organization management |
| 23 | 7 | 0.998 | 2007 | Construction professional | Project management |

### 3.2.2. General Research Themes Derived from Clusters

Based on the label generated by *CiteSpace* and by reviewing the publications in the above 15 clusters, seven major themes of articles related to construction management between 2000 and 2020 are manually summarized below.

(1)　Theme 1: Modern technology

Although the *CiteSpace* automatically labeled cluster #0 as "Construction project management", and cluster #11 as "Visualization software", after reviewing articles in these two clusters, it was found that these articles were mainly related to modern technologies used in

construction management. In cluster #0, the top cited article mentioned the importance of adopting the information and communication technologies (ICT) in construction industry as it is extremely dependent on the access and management of data [40]. Various methods were introduced on managing key issues with construction management such as costs, planning, risks, safety, progress monitoring, and quality control with the help of Building Information Modeling (BIM) and ICTs [41,42]. Similarly, Papadonikolaki (2018) introduced the BIM adoption in Dutch construction [43]. Ma et al. (2018) established a conceptual framework to integrate BIM into life-cycle project management [44]. The most citing articles provided a basic knowledge of BIM adoption in construction projects, including its trends, benefits, risks, and challenges [45], BIM handbook for practice [46], and pressures on BIM adoption [47]. As for cluster #11, the most highly cited articles by Chau et al. (2005) and Ma et al. (2005) both discussed the application of 4D for dynamic site layout with a prototype four-dimensional site management model (4DSMM) and a 4D Integrated Site Planning System (4D-ISPS), respectively [48,49]. As for the highly citing articles, Li et al. (2018) developed a learning tool combining lean construction principles and information technologies, which are radio frequency identification (RFID) and BIM [50]. Therefore, these clusters summarized the contributions and benefits of the adoption and integration of BIM and other technologies into the construction industry and provides insights into better diffusion of ICTs in the construction industry.

(2)    Theme 2: Waste management

Cluster #1, labeled as "Construction waste management", and Cluster #3, labeled as "Demolition waste management", both focused on the waste management on construction sites. The top cited article in Cluster #1 established a framework to help understand the construction and demolition waste management (C&D WM), and found three major topics, which are C&D generation, reduction, and recycling [51]. Based on this and in order to achieve efficient C&D waste management, another two articles were also highly cited. Chen et al. (2019) gave a clear description of the decision-making behaviors of major participants in C&D WM [52], and Nikmehr et al. (2017) explored the factors influencing the management of C&D waste in Iran [53]. Similarly, top cited articles in Cluster #3 also studied the C&D WM from the perspective of sustainable development [54], stakeholder behavior [55], and the policy making process [56], caring about environment protection and cost-efficient recycling. Therefore, during the past two decades, C&D WM has obtained more and more attention. The articles in these clusters gave a better understanding of the C&D WM framework, explored many sustainable methods for resource recycling, and considered the influence of human behaviors for efficient waste management with game theories.

(3)    Theme 3: Performance management

Cluster #2, labeled as "exploring critical success factor", and Cluster #6, labeled as "Critical success factor" are both under the theme of performance management. For Cluster #2, project performance influencing factors have gained the most attention in both cited and citing articles, such as time, cost, quality, and other procurement-related factors [57,58]. Studies in this cluster found that accurate cost estimates and program budgets, as well as a clear understanding of the causes and effects of schedule delays were critical for project performance management, especially in transportation projects [59,60]. As for cluster #6, Wang et al.(2020) investigated how financial performance could be enhanced by internationalization, and found that both economic and social risks play a positive moderating role in the relationship between internationalization and financial performance [61]. Based on the highly citing article by Anantatmula (2015), Ecem Yildiz et al. (2020) further studied the factors that could enhance project performance and simulated dynamic relationships between popular performance measures and planning strategies with the system dynamic (SD) method [62,63]. Besides, Lee and Han (2017) and Zhao et al. (2017) both conducted research on international project performance [64,65]. In summary, different factors such as planning strategies, socio-economic factors, design and maintenance, and international

contractor properties and their impacts on performance management were studied in these clusters.

(4)    Theme 4: Risk management

Cluster #4, labeled as "Enterprise risk management", and Cluster #9, labeled as "risk management", are under the theme of risk management. The highly cited articles in Cluster #4 mainly focused on the enterprise risk management (ERM) in indoor and international construction firms. Zhao et al. (2014) investigated the ERM maturity in Chinese construction with 35 Chinese construction firms and found three key risk factors: communication, objective setting, and risk-aware culture [66]. Later, they also developed a knowledge-based decision support system for enterprise risk management (KBDSS-ERM) for Chinese construction firms to facilitate their ERM implementation [67]. The highly citing articles are not limited to the construction field. For example, Doherty and Smetters (2005) identified moral hazards in the traditional reinsurance market [68]. Breuer (2005) explored optimal contracts under multiple moral hazards to monitor and control losses and insurer risks [69]. In addition to discussing ERM and the usage of construction insurance in dealing with risk, cluster #9 included more discussion on different stakeholders' risks under various situations. For example, Rockart and Lecturer (2000) presented a risk assessment model to assist in evaluating potential risks in international construction projects [70]. Wang and Chou (2003) studied risk allocation by contract clauses and gave suggestions on contractor's risk handling strategies [71]. Chan et al. (2011) identified the three most important risk factors for PPP projects in China, which were government intervention, government corruption, and poor public decision-making processes [72]. Similarly, in the highly citing articles, Bing et al. (2005) and Li et al. (2005) discussed the risk allocation of PPP projects in the UK and suggested that more efficient risk allocation frameworks should be established by the public sector in the early stages of project development [73,74]. Some other highly citing articles also highlighted the importance of stakeholder inputs in dealing with project failure risks. Studies in these two clusters emphasized that suitable risk analysis and management were essential for the overall success of construction projects.

(5)    Theme 5: Project management

Cluster #5, labeled as "Civil engineering", and Cluster #23, labeled as "Construction professional", consider the issues related to construction project management. In the problems studied in Cluster #5, changes of design during construction, cost overruns, delay, competitive tendering procedures, and late payment were the five most critical project management challenges identified [75], and delay is the most significant one among them. Specifically, Gunduz et al. (2015) built a decision support tool to quantify the probability of delay in construction projects in Turkey with 83 identified delay factors by a fuzzy-relative importance index [76]. Khair et al. (2018) addressed the key delay factors in the road construction industry in Sudan and proposed a framework for minimizing and controlling delays in construction projects [77]. Articles in Cluster #5 and #23 also discussed various topics under the theme of construction project management. The highly cited articles developed a BIM-project information management framework for construction project management [78], explored the success factors of effective project management with structural equation modeling [79], and issues associated with design and construction industry globalization [80]. The research outcomes from these studies can help to decrease project delays and cost overruns, provide guidance for construction labor productivity in theory and practice, and also provide proof that sustained interactions between culturally and linguistically diverse networks may lead to multicultural networks which outperform mono-cultural networks, which is consistent with the research findings of Di Marco et al [81]. It can be concluded that Cluster #5 and #23 discussed the major problems faced within project management and provided possible solutions for further improvement on multicultural projects.

(6)     Theme 6: Knowledge management

Cluster #7, labeled as "knowledge management practice", discussed issues related to knowledge management for engineering and construction. As knowledge management has quickly become a key organizational capability for the construction industry to create competitive advantage, the top cited article proposed a model for benchmarking the knowledge management performance of construction firms with a fuzzy-weighted average algorithm [82]. Besides construction firms [83], Hwang and Ng (2013) built a knowledge base for project managers who execute green construction projects and to identify the key knowledge areas and skills. Construction knowledge can be shared by using BIM technology [84], ontology, and semantic web technology, also offering an opportunity to enable knowledge such as risk knowledge in the domain of safety management to be represented semantically [85]. Therefore, knowledge management discussed in this cluster is performed on the level of construction project, company, organization, and person [86–88], and also serves other themes [89], such as risk management (Theme 4), performance management (Theme 3), and organization management (Theme 7) with modern technology (Theme 1).

(7)     Theme 7: Organization management

Cluster #19, labeled as "Chinese construction firm", focused on issues related to organization management in the construction industry. Based on the overview of megaprojects given by Flyvbjerg (2014) as the highly citing article, participating organizations' relational behavior [90,91], expectations, and satisfaction among the stakeholders [92], and organizational citizenship behaviors for environment [93] were examined for megaprojects among the most cited articles. Various factors were identified for improving the relationship quality in megaprojects based on related theories such as the theory of planned behavior (TPB) and stakeholder theory [91]. Some highly cited articles also discussed the issues related to relationships in project organizations, mainly based on social network analysis, such as examining the relationship between project performance and organizational characteristics in construction companies [94], predicting changes in market leadership [95], and exploring dynamics in global construction industry [96]. The most citing articles are related to common theories about organizational networks and collaborations [97,98]. The changes in the engineering and construction industry in the 21st century require organizations to play a more active role in developing knowledge management and organizational learning initiatives with modern technology [99,100]. In total, this cluster dealt with the cooperation behavior, network relationships among stakeholders, and organizational knowledge sharing in megaprojects under the environment of global construction.

*3.3. Research Trend*

3.3.1. Keyword Analysis

The top 10 frequently used author's keywords (DE) and keywords plus (ID) in each period are shown in Table 5. The frequency (F) in the table means the count of the keyword occurrence. Some differences can be found between ID and DE during the period. The DE in each period appeared in the clusters of every co-citation analysis which illustrate the general research theme. While ID is much more related to some basic concepts. It can be found that in Period 1, basic concepts such as system and model were introduced and discussed. In Period 2, performance got the most attention and the related management concepts and models studied in Period 1 were adopted in the practice. In the total period, performance and model were the most frequently discussed keywords in the field of construction management, and some modern technologies and ideas such as BIM, simulation, circular economy, and lean construction were adopted for the case study, further innovation, framework establishment, and strategy building in engineering.

**Table 5.** Top 10 frequencies of keywords in each period.

| Period 1 (2000–2009) | | | | Period 2 (2010–2020) | | | |
|---|---|---|---|---|---|---|---|
| **ID** | **F** | **DE** | **F** | **ID** | **F** | **DE** | **F** |
| system | 10 | construction | 24 | performance | 133 | construction | 103 |
| model | 9 | construction management | 24 | model | 102 | management | 85 |
| design | 7 | construction industry | 20 | system | 85 | construction management | 74 |
| performance | 6 | management | 15 | design | 65 | project management | 58 |
| industry | 5 | project management | 11 | impact | 52 | construction industry | 53 |
| systems | 5 | knowledge management | 10 | projects | 51 | construction and demolition waste | 44 |
| impact | 4 | information management | 7 | framework | 49 | risk management | 41 |
| quality | 4 | information technology | 6 | industry | 48 | waste management | 29 |
| behavior | 3 | performance | 5 | systems | 47 | project | 26 |
| health | 3 | safety | 5 | behavior | 26 | sustainability | 21 |

Note: ID means keywords plus; DE means author's keywords; F means frequency.

### 3.3.2. Research Trend Evolution

In order to understand the evolution of research trends in the construction management field, three strategic diagrams (Figure 7) were developed with *Bibliometrix R* by keyword co-occurrence analysis. The keyword used in the analysis is ID, as the quality of these terms are more significant than DE [20].

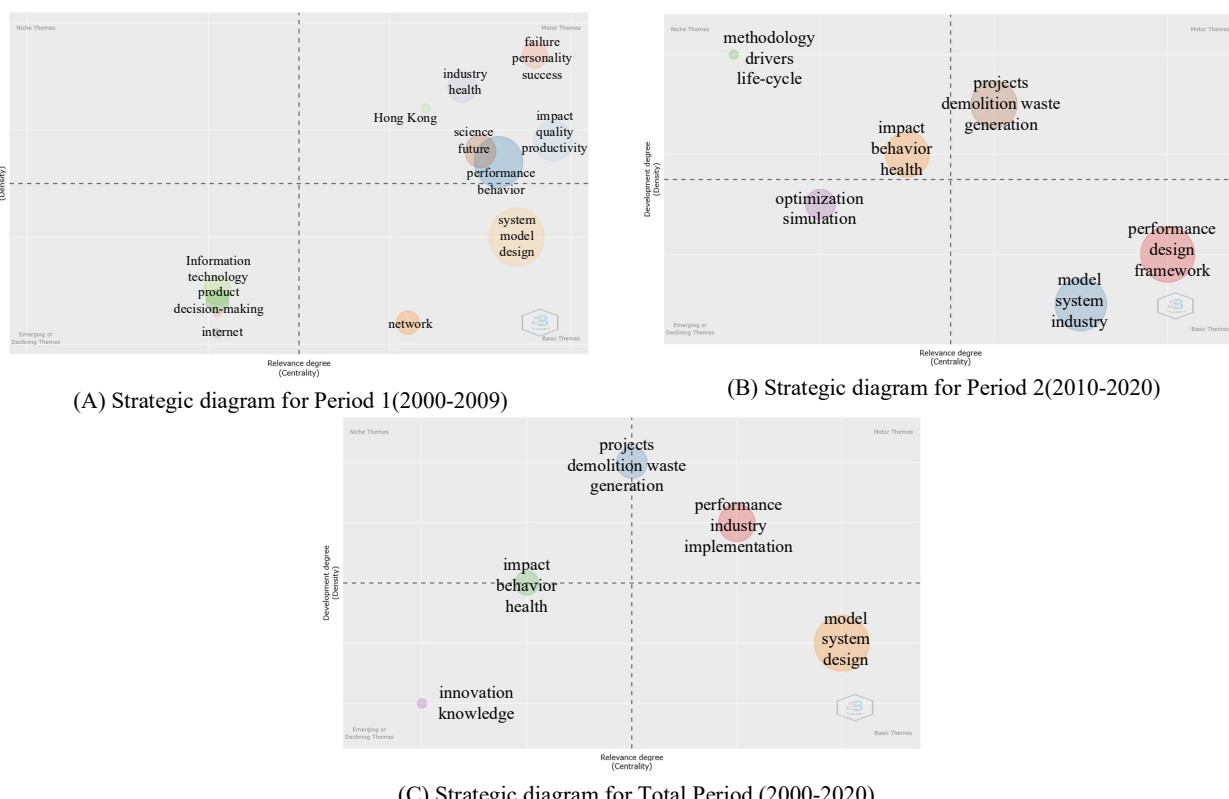

(A) Strategic diagram for Period 1(2000-2009)

(B) Strategic diagram for Period 2(2010-2020)

(C) Strategic diagram for Total Period (2000-2020)

**Figure 7.** Strategic diagrams for each period (generated by *Bibliometrix R*).

With regards to (Figure 7A), from 2000–2009, various research themes were more fully-developed, such as *failure, industry, impact, science, and performance*. These clusters and their related topics are supported by the research performed in the early 21st century or even prior, such as the studies related to failure management in construction industry by Malin and Lance in 1987 [101]. Studies related to *information, technology and internet* began to emerge and get attention in this period, such as the adoption of mobile information

and communication technology [102] and RFID technology [103] in construction industry. Studies in *model, system, and design* became the basic study concept in this field, and continued to Period 2.

In Period 2 (Figure 7B), the trend evolved. Studies related to *projects, demolition waste and generation* became the most remarkable research trend, and this is accordance with the general themes identified in Section 3.2.2. Studies related to *optimization and simulation* become new emerging trends in this field. Various models in major themes such as demolition management and project management were built based on optimization or simulation [104,105]. Studies related to *model and system* remain the basic concept in the area.

In the total period (Figure 7C), studies related to *performance* become the most developed trend in the field as discussed in Theme 3 (Section 3.2.2). The term *model and system* remained the basic theme. The emerging theme changed to *innovation and knowledge*. Studies in *impact, behavior and health* started from Period 2 and hoped to become motor themes in the future. Therefore, compared with the themes identified by co-citation analysis, the theme of *performance* (Theme 3) obtained the most attention during the whole period. Themes such as waste management (Theme 2) and project management (Theme 5) are common topics in the field of construction management with long term influence. The theme of modern technology (Theme 1) appeared in both the two periods with different emerging terms continued to evolve over the period. All the themes are closely linked and have influences on each other.

## 4. Integrative Analysis Results and Discussion

Based on the above bibliometric analysis, details about the research data, data-driven approaches, and directions in the field of construction management are discussed in this section.

### 4.1. Types of Research Data

In addition to standard information such as finance reports, demand from customers, and operating records, a lot of unique data based on the specific characteristics of a certain type of project are needed to better understand and study them. This, therefore, poses a challenge for the research related to engineering management. Traditionally, data from questionnaires, interviews, and historical records are commonly used for qualitative and some quantitative research [106,107]. However, as these data are stored in different formats, in some cases, the required historical data may be insufficient or even deficient [108]. As shown in the research themes, megaprojects and international construction cooperation are becoming more and more popular recently. Besides, with the advancement of data acquisition technologies, a large amount of new data is required for better analysis of related issues, such as social media data.

With the development of ICTs, the types of data used for research have changed a lot. The construction project management processes are transitioning from two-dimensional paper-based to three-dimensional digital data-based during the entire life cycle with the help of BIM and other technologies. Large sets of data can be obtained and various simulation methods or models can be adopted based on the large data collected. Besides, innovative data sources are used in recent studies, which increases the variety of data type. For example, human opinions or perceptions captured from texts, pictures, and videos collected from job sites and social media platforms have been used in studying construction problems [109]. Sensors and eye-tracking devices to capture human attention and cognition in construction are also equipped to ensure the safety in the construction site [110].

### 4.2. Data-Driven Approaches

Qualitative analysis is commonly used in project management studies since the 1990s, and the qualitative data covers all text-based studies. The data can be first-hand data collected from interviews and investigations, as well as the second-hand data mainly

collected from literature, press publications, social media, and historical documents. For example, Barco (1994) compared various maintenance and repair-budget models from investigation that can be applied to develop a maintenance and repair financing plan at an appropriate level [111]. Antillon et al. (2018) analyzed the data from interviews and project documentation to explore lifecycle decision-making processes for each project with the help of the qualitative analysis tool *NVivo* [112].

On the other side, quantifiable data were gathered and statistical, mathematical, or computational techniques were used to perform quantitative analysis. Three popular quantitative analysis approaches were mostly used in the reviewed articles. First is survey research. For example, in the theme of organization management (Theme 7), a survey is one of the most commonly used methods. The surveys can be conducted through phone, email, mail, online, or face-to-face interview, collecting answers from micro, meso, or macro levels [113,114].

Second is correlational research. Usually, it is used to correlate two or more variables using methods of mathematical analysis, such as structural equation modeling or regression analysis. The relationships between critical project success factors or barriers with residual value risks, delays, disputes, safety, and anticorruption in infrastructure projects were explored and discussed by many researchers in the related clusters [115–117].

Last is the simulation method. As the new trend discovered in Section 3.3.2, simulation modeling makes it possible to compare different types of failures, recovery strategies, and resource allocations to provide more details of the behaviors of a system under different scenarios with the help of modern technologies [108]. Common simulation techniques such as system dynamic are used in construction management research, such as delay risk effects investigation [118], conflict management [119], and contractual relationship modeling [120]. Other techniques such as agent-based modeling are also popular in investigating information sharing strategies, safety behaviors, and solving goal incongruence problems in construction projects [121,122].

### 4.3. Future Research Directions

In addition to further studying the traditional topics in the construction management field, such as organization, risk, and performance, there is a need to consider the changes to the domain brought by characteristics of the data-rich era such as megaprojects, globalization, and digitizing. Some new directions are suggested to be further studied considering emerging topics, such as the modern technology, sustainability, climate change, and human development. The explanation for each direction is as follows.

### 4.3.1. Smart Construction

Based on the research themes and trend analysis, modern technologies such as BIM and other ICTs are expected to obtain further adoption and development in the field of construction management. With the development of smart cities, the adoption of intelligent methods should be explored for further construction. Therefore, it is important to find ways to incorporate smart knowledge and technologies into the construction of smart cities, such as intelligent traffic control and smart construction site management [123]. Besides, when considering risk management for the construction industry, threat identification and monitoring can be conducted by computers and help to mitigate human-related risks. New technologies and concepts such as digital twins, machine learning, and automation construction are promising for future engineering studies [124,125]. Examples of important future research and practice questions in this direction include how to popularize 3D-printing technique in construction site work, how to optimize the use of robots for construction in a cost-effective way, and how to establish digital twins that accelerate and automate the traditional design, construction, and operation processes.

### 4.3.2. Sustainability-Based Development

Sustainability has always been an important direction for future development. The current identified research themes such as organization management, performance management, and waste management are proven to be common topics with long term influence on sustainability. Articles under these themes are devoted to providing services and creating a better and sustainable life for human beings. With the rapid speed of urbanization, the increasing demands on urban infrastructures require that city officials find new ways of keeping infrastructures in good condition, as well as engaging people in a way that promotes efficient and sustainable usage of them from physical, environmental, economic, and social perspectives [126,127]. Therefore, it is important to build a comprehensive framework for further studies on economic, social, and environmental sustainability issues. Potential future research questions in this direction include how to reduce on-site and off-site pollution with lean management, how to promote and regulate the global carbon trading market, and how to monitor the mental and physical health of older construction workers. There is also a pressing need to take the impact of government supports, public and private engagement, and legal and regulatory environment on sustainability issues into consideration.

### 4.3.3. Resilience-Based Planning and Management

Although cluster labels identified did not explicitly include resilience, some recent articles addressed the importance of developing resilience in various kinds of infrastructures to cope with different disruptions such as global climate change. Considering the frequent natural hazards such as floods and hurricanes, as well as the world pandemic COVID-19 in 2020, construction and infrastructure projects are expected to continually face complex and uncertain environments in the future. Although some approaches have been proposed to assess the costs and benefits of adapting infrastructure to disruptions such as climate change and natural disasters [128,129] or COVID-19 [130], more studies are needed to investigate the resistance, adaptation, and resilience of engineering projects, companies, and organizations with supported data. Issues related to the interdependence of networked critical infrastructures, the resilience of construction companies and workers against pandemics, and the management of reconstruction duration, cost, sequence, and resources after natural disasters might become the focus of future studies in this direction.

### 4.3.4. Human-Centric Consideration

During the analysis, it is found in many clusters that the role played by human and organizational factors on the performance of construction projects has obtained more and more attention [131,132]. As humans play an essential role in any engineering work, it is necessary to examine the influence of human-related factors, such as leadership, cooperation, and learning, on project performance. In addition, more attention on human-centric considerations (e.g., public concern, social equity) is suggested to be paid for project assessment during the life-cycle of different kinds of construction in practice. Psychological factors, such as attitudes, biases, and behaviors of different stakeholders, are suggested to be further investigated in future studies in order to achieve higher levels of social benefit and public satisfaction.

### 4.4. Challenges towards Data-Driven Research

Considering all the existing research and future research directions above, data is the basis of all research, whether qualitative or quantitative. Articles in the field of construction management fall into the category of engineering and require sufficient data. Fortunately, the proliferation of big data provides a wide range of opportunities for data-based engineering management, and many studies have enjoyed its convenience [123]. However, this also poses challenges for current studies on data collection and information transfer. In construction projects, data could be stored by different stakeholders and are difficult to obtain due to confidentiality or other reasons. Therefore, the breakthrough challenge

involves exploring a uniform method of valuable data mining and analysis, as well as understanding how to use the data to make the life-cycle management of construction projects more effective and sustainable [108]. It is suggested to further research the co-ordination of data collection, access, and sharing among different parties, especially in engineering projects. It is also suggested to explore innovative approaches (e.g., blockchain technology and smart contract) to ensure the data quality and privacy through better data management, analytics, integration, and protection in the future [133]. In addition, it will be more valuable to use data in applications such as autonomous driving, government decision-making, military command, and medical health, especially in the fields that are closely related to human life, property, development, and security. There still face a series of major fundamental theoretical and core technical challenges to be solved if the data are to be effectively applied.

## 5. Conclusions

With entering the data-rich era, this paper provides an overview of the articles published in the field of construction management from 2000 to 2020. Based on the number of articles published annually, two different development stages (i.e., 2000 to 2009 as Period 1, 2010 to 2020 as Period 2) were recognized. Seven major research themes in the field were identified through bibliometric analysis, namely modern technology, waste management, performance management, risk management, project management, knowledge management, and organization management. The kind of research data and data-driven approaches were discussed with the above seven research themes. It was found that research data types and approaches have evolved with the development of information technologies. Also, future research directions in smart technology, sustainability, resilience, and human factors are suggested for academics to continue working on in the field of construction and engineering management, connecting current research progresses to future challenges and opportunities towards a data-rich era.

Solving construction management related issues not only enables economic development, job creation, and the delivery of local goods and services, but also enhances quality of life for citizens, helps protect natural resources and environment, and promotes a more effective and efficient use of financial resources. Such research needs the support of large sets of various kinds of data. There are large amounts of existing studies based on the data collected traditionally from questionnaires, interviews, and historical records. It helps to have a better recognition of the current situation with evaluations, comparisons, and mechanism explorations. With entering into the era of information and big data, new kinds of data as well as the data-driven approaches have adopted and achieved some good applications in the identified research themes. However, the result is still far from our expectations, and big data application is still in its infancy. It is not enough to summarize and extract relevant information and knowledge from big data, but also necessary to predict the development trend, and guide and optimize the decisions accordingly based on the relationship and development model analyzed from big data. In the future, with the expansion of application fields, the improvement of technology, the open mechanisms for data sharing, and the maturity of the industrial ecology, predictive and guiding applications with greater potential value will be the focus of development in the domain.

The review and discussion in this paper provides an opportunity for researchers and practitioners to reflect what has been done and what needs to be done in the future of the construction management domain with the fast development of information and big data. Researchers can use the future research directions identified to guide their research map, as well as mining the value of data. The limitation of the paper is that the results are more helpful to academia, and the theoretical significance of the study is greater than the practical contribution. More practical cases will be required in the future to assist practitioners such as construction companies and decision-makers in better understanding the frontier of technology and management theory and applying it to their daily work. Thus, both researchers and practitioners can gain a more comprehensive and systematic

knowledge in the field of engineering management and continue to further explore certain areas to address the grand challenges with better use of various kinds of data.

**Author Contributions:** Conceptualization, S.Z. and D.L.; methodology, S.Z.; software, S.Z.; validation, J.Z. and H.F.; formal analysis, S.Z.; data curation, H.F.; writing—original draft preparation, S.Z.; writing—review and editing, D.L. and J.Z.; visualization, S.Z.; supervision, D.L. and H.F.; funding acquisition, S.Z. and D.L. All authors have read and agreed to the published version of the manuscript.

**Funding:** This project is funded by the National Natural Science Foundation of China (No.72204127) and National Social Science Fund of China (No. 19BGL281).

**Data Availability Statement:** Not applicable.

**Conflicts of Interest:** The authors declare no conflict of interest.

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
