# Peer review of "Towards a Data-Rich Era: A Bibliometric Analysis of Construction Management from 2000 to 2020"

_buildings, doi:10.3390/buildings12122242_

Round 1
Reviewer 1 Report
I appreciate the authors’ efforts in the work related to a bibliometric analysis of construction management from 2000 to 2020. The information is relatively easy to navigate, and the structure of the paper allows readers to analyze the concepts approached.
However, authors must improve the article by paying attention to the following issues:
(1) The introduction should clearly state the research question(s) or the research objective(s) that the authors present for this paper.
(2) The literature review section is absent and it must provide a good background of the domain. The authors should relevant and interesting arguments to the investigated field.
(3) The reference could be enhanced with some updated works, such as:
- Exploring the Research Regarding Frugal Innovation and Business Sustainability through Bibliometric Analysis. Sustainability. 2022, 14(3), 1326. https://doi.org/10.3390/su14031326.
- Bibliometric Analysis of the Green Deal Policies in the Food Chain. Amfiteatru Econ. 2022, 24, 410–428. DOI:10.24818/EA/2022/60/410.
(4) It is unclear why the study period is restricted to 2000 - 2020. This aspect needs a more elaborate justification.
(5) Authors need to explain about study limitations and as well in revised draft.
Reviewer 2 Report
REVIEW COMMENTS
After the review process, the reviewer would like to give some critical thinking and idea to help authors get their job done efficiently.
I have only a few concerns about the paper and some suggestions that maybe the authors could consider:
1. To begin with, there are some typos and grammar mistakes. Some long sentences could make readers confused.
2. In the 'Introduction' section, the proposed research gap and the stated objectives do not meet the criteria of proper synergy. Please make the research gap and the research objectives consistent with each other.
3. I think that the “Introduction” section can be improved by adding a new bibliometric analysis of articles from different fields to highlight the importance of bibliometric analysis, which become widely use recently, I suggest some references which can be beneficial for this, i.e., 'a global research trends of neuromarketing: 2015-2020'
4. It might be appropriate for the authors to explain why they had chosen the period of extracting data between 2000 and 2020?
5. Why did you exclude the papers from the years 2021 and 2022?
6. Why have you chosen these specific keywords?
7. Could the authors explain why they only focused on the English articles?
8. Could the authors clarify why they focused only on articles and what type of documents have been excluded?
9. Although the development of search criteria does not justify why the decisions are made (for example, why it is searched in WOS and not in Scopus or both altogether).
10. In section 2.2 Bibliometric analysis, I can suggest a reference which can improve this section, i.e., doi.org/10.1080/23311975.2021.1978620.
11. Could the authors explain why they have used R-tool package rather than VOSviewer?
12. The authors should explicitly state the novel contribution of this work and its similarities and differences with their previous publications.
13. The authors need to clearly articulate the implications of the research results for theory and practice are not included in the article. A detailed explanations of the author’s recommendations should be included. I would suggest writing a paragraph in the conclusion section for the implications. I would suggest the ref. which can help in this issue ‘neuroimaging techniques in advertising research: main applications, development, and brain regions and processes’. Also, state a few of the key implications at the end of the 'Introduction' section.
14. The authors need to clearly articulate the limitations and Future research directions should be proposed. I would suggest the ref. 'current trends in the application of eeg in neuromarketing: a bibliometric analysis' which can help in that issue.
15. For readers to quickly catch your contributions, it would be better to highlight major difficulties and challenges and your original achievements to overcome them in a clearer way in the abstract and introduction.
16. How could/should your study help the future studies?
If these revisions can be made in the manuscript, I believe that this study can be accepted for publication.
I wish the authors all the very best with this study.
Round 2
Reviewer 2 Report
REVIEW COMMENTS
After the review process, the reviewer would like to give some critical thinking and idea to help authors get their job done efficiently.
I have only a few concerns about the paper and some suggestions that maybe the authors could consider:
1. I observed some references added in the reviewers' responses draft, but they are missing in the revised copy of the paper. For example, No. 12 in the reference list in your reviewers' responses draft not included in the manuscript. The authors should add the following reference which is in the reviewers responses draft to the original of the manuscript.
12. Alsharif, A.H.; Salleh, N.Z.M.; PilelienÄ—, L.; Abbas, A.F.; Ali, J. Current Trends in the Application of EEG in Neuromarketing: A Bibliometric Analysis. Sci. Ann. Econ. Bus. 2022, 69, 393–415, doi:10.47743/saeb-2022-0020.
2. Could the authors clarify why they focused only on articles and what type of documents have been excluded? The authors should write the type of documents that have been excluded in the original article, not only in response to the reviewer draft. Therefore, making the extracting process of articles more clear for normal readers.
3. I think that the "Introduction" section can be improved by adding a new bibliometric analysis of articles from different fields to highlight the importance of bibliometric analysis, which become widely use recently, I suggest some references which can be beneficial for this, i.e., 'neuroimaging techniques in advertising research: main applications, development, and brain regions and processes'
4. The authors need to clearly articulate the limitations and Future research directions should be proposed. I would suggest the ref. 'neuroimaging techniques in advertising research: main applications, development, and brain regions and processes' which can help in that issue.
5. For readers to quickly catch your contributions, it would be better to highlight major difficulties and challenges and your original achievements to overcome them in a clearer way in the abstract and introduction.
If these revisions can be made in the manuscript, I believe that this study can be accepted for publication.
I wish the authors all the very best with this study.
Round 3
Reviewer 2 Report
All of my concerns have been fully addressed in the revised version.
Best wishes